# Improving Sperm Oxidative Stress and Embryo Quality in Advanced Paternal Age Using Idebenone In Vitro—A Proof-of-Concept Study

**DOI:** 10.3390/antiox10071079

**Published:** 2021-07-05

**Authors:** Victoria Nikitaras, Deirdre Zander-Fox, Nicole O. McPherson

**Affiliations:** 1Robinson Research Institute, University of Adelaide, Adelaide, SA 5005, Australia; victoria.nikitaras@gmail.com; 2Adelaide Health and Medical School, University of Adelaide, Adelaide, SA 5005, Australia; dzander@monashivfgroup.com; 3Repromed, Dulwich, SA 5065, Australia; 4Department of Bioengineering, University of South Australia, Adelaide, SA 5072, Australia; 5Monash University, Clayton, VIC 3800, Australia; 6Monash IVF Group, Richmond, VIC 3168, Australia; 7Freemasons Centre for Male Health and Wellbeing, University of Adelaide, Adelaide, SA 5005, Australia

**Keywords:** reactive oxygen species, male infertility, in vitro fertilization, assisted reproductive technology, male reproduction

## Abstract

Advanced paternal age is associated with increased sperm reactive oxygen species (ROS) and decreased fertilization and pregnancy rates. Sperm washing during infertility treatment provides an opportunity to reduce high sperm ROS concentrations associated with advanced paternal age through the addition of idebenone. Sperm from men aged >40 years and older CBAF1 mice (12–18 months), were treated with 5 µM and 50 µM of idebenone and intracellular and superoxide ROS concentrations assessed. Following in vitro fertilization (IVF), embryo development, blastocyst differentiation, DNA damage and cryosurvival, pregnancy and implantation rates and fetal and placental weights were assessed. Five µM of idebenone given to aged human and mouse sperm reduced superoxide concentrations ~20% (*p* < 0.05), while both 5 and 50 µM reduced sperm intracellular ROS concentrations in mice ~30% (*p* < 0.05). Following IVF, 5 µM of idebenone to aged sperm increased fertilization rates (65% vs. 60%, *p* < 0.05), blastocyst total, trophectoderm and inner cell mass cell numbers (73 vs. 66, 53 vs. 47 and 27 vs. 24, respectively, *p* < 0.01). Treatment with idebenone also increased blastocyst cryosurvival rates (96% vs. 78%, *p* < 0.01) and implantation rates following embryo transfer (35% vs. 18%, *p* < 0.01). Placental weights were smaller (107 mg vs. 138 mg, *p* < 0.05), resulting in a larger fetal to placental weight ratio (8.3 vs. 6.3, *p* = 0.07) after sperm idebenone treatment. Increased sperm ROS concentrations associated with advanced paternal age are reduced with the addition of idebenone in vitro, and are associated with improved fertilization rates, embryo quality and implantation rates after IVF.

## 1. Introduction

The average age of first-time fathers is increasing worldwide. In the past 60 years in Australia, the median age of fathers has increased from 29.0 years to now 33.1 years (~15%) [1], with 10% of all naturally conceived births now coming from fathers ≥40 years [2]. These statistics are heightened within couples seeking assisted reproductive technology (ART) treatment, due to the strong relationship between paternal age and sub-fertility [3,4,5,6]. At present, the average age of men undergoing ART in Australia is 38.1 years, with one third of all initiated cycles in men aged ≥40 years [7]. While it is well known that advance maternal age is associated with reduced fertility and a wide variety of reproductive complications [8], it is only now becoming apparent that advanced paternal age is also associated with a reduced ability to conceive [3,4,5,6] due to its negative associations with sperm parameters (count, motility and morphology) [9] and reproductive hormone levels [10].

More recently, advanced paternal age has been associated with higher levels of semen ROS concentrations and associated sperm DNA damage [11,12,13,14]. Oxidative stress is characterized by the overproduction of reactive oxygen species (ROS) including superoxide anions, peroxides and hydroxyl radicals, which outweigh the ROS neutralizing capacity of cellular antioxidant systems [15,16]. Increased oxidative stress results in lipid peroxidation, loss of motility and reduced sperm–oocyte binding [17], along with DNA damage which has been shown to be inherited by the preimplantation embryo and can have negative consequences on pregnancy and offspring health [18]. Sperm require ROS for normal cellular functioning including triggering the signaling cascade for stimulating capacitation, which is a pre-requisite for fertilization [19]. However, as sperm lose the majority of their antioxidant capabilities through cytoplasmic shedding during spermiogenesis, it leaves them vulnerable to oxidative attack [20]. Due to this relative lack of antioxidant protection, sperm are heavily reliant on the surrounding fluid for protection. Seminal plasma contains some of the most highly specialized antioxidants and scavenging enzymes known including glutathione peroxidase (GPx5), extracellular superoxide dismutase (SOD), uric acid, vitamin C, tyrosine and polyphenols [21,22]. However, in standard ART protocol, sperm are washed free of their seminal plasma components to reduce exposure to proteins that inhibit sperm motility and reduce sperm viability (i.e., semenogelin I and II) [23,24]. The seminal plasma component is replaced with a culture medium containing carbohydrates, amino acids, buffers, antibiotics and human serum albumin (HSA) with minimal antioxidant properties (i.e., some media contain ETDA and taurine; however, the antioxidant capacity of this is low). The process of sperm washing, therefore, completely removes nearly all the sperm-extrinsic antioxidant components, rendering sperm defenseless to oxidative damage. As sperm are proficient generators of ROS, they are only left with their poorly functioning antioxidant defenses (SOD, GPX and catalase) to combat oxidative damage [25,26]. This, coupled with aging, which is already associated with its own reduced antioxidant defenses [27], adds further insult to the already poorly defended sperm.

Antioxidant supplements have been explored as a potential treatment for male infertility, both in vitro and in vivo, as at a cellular level they scavenge ROS and decrease oxidative stress [28,29,30,31,32,33,34,35]. However, very few studies have assessed their benefits in the context of advanced paternal age. A cross-sectional study assessing dietary vitamin C, E and zinc intake in men aged >44 years found a 20% reduction in sperm DNA damage in those men who had the highest micronutrient intake compared with those with the lowest [36], indicating that dietary antioxidants may be beneficial for reducing sperm oxidative stress associated with advanced paternal age. In terms of in vitro antioxidant use, there have been a number of studies assessing whether or not the addition of antioxidants to sperm culture media during human sperm preparation could be beneficial to sperm function [31,37,38,39,40], although these studies have not assessed their outcomes in the context of advanced paternal age. While, these studies have shown some beneficial effects to sperm motility, sperm ROS production, lipid peroxidation and sperm viability with the addition of EDTA, lycopene, zinc, ascorbic acid (vitamin C), coenzyme Q10, taurine and glutathione to the medium, little has changed in the composition of current commercial sperm culture medium. One of the biggest limitations of the studies to date is the addition of antioxidants that are not cell-permeable, such as vitamin E and coenzyme Q10, which are instead lipophilic. These tend to be retained in cell membranes and only act via ROS scavenging [41]. Therefore, the positive effects seen in these studies maybe more due to extrinsic ROS removal via antioxidant scavenging as opposed to reducing sperm ROS generation and intrinsic ROS scavenging. Given the fatty acid-rich plasma membrane of sperm, it is vital that the antioxidants added to sperm culture medium are cell-permeable and are capable of reaching the relevant sites of free radical generation such as the mitochondria, which is the primary source of ROS production in sperm [25]. Furthermore, few in vitro and in vivo studies have examined the downstream effects on embryo development and pregnancy viability [28,29,34,42], which are required in order to provide an evidence base for prescription and use of antioxidants in clinical practice.

Idebenone is a mitochondrial-permeable synthetic benzoquinone which functions as an antioxidant by scavenging free electrons, as well as acting as a proton and electron carrier in the mitochondrial transport chain, increasing energy production [43,44,45]. More recently, idebenone has been shown to display molecular activity outside of an antioxidant including bioactivity of idebenone metabolites, protein inhibition (i.e., p52Shc), regulation of gene transcription (i.e., *Lin24A*) and reductions in inflammation and endoplasmic reticulum stress [44]. It shares similarities with the antioxidant ubiquinone, known as co-enzyme Q10; however, it is more readily soluble in gamete/embryo-compatible culture media solutions, is able to permeate cellular membranes in vitro and is already been approved by the Therapeutics Goods Administrator (TGA) and US Food and Drug Administration (FDA) to treat Duchenne muscular dystrophy [44], thus making it a good candidate for use in clinical ART. In vitro, idebenone has already been shown to reduce ROS concentrations and cell death in retinal epithelium [46], lipid peroxidation of vascular endothelial cells [47], reduce ROS formation in rat brain synaptosomes [48] and apoptotic cell death in optic nerve astrocytes, all which are hallmark features of oxidative damage and common phenotypes seen in sperm from men of an advanced aged [49]. Therefore, the aim of this proof-of-concept study was to determine if the addition of a membrane permeable antioxidant (idebenone) to sperm culture media could reduce sperm ROS concentrations associated with advanced paternal age, and define its subsequent effects on fertilization, embryo development and implantation rates utilizing a mouse model of advanced paternal age.

## 2. Materials and Methods

### 2.1. Humans and Ethic Approval

Surplus normozoospermic semen samples (defined by sperm concentration of ≥15 × 10^6^/mL and ≥39 × 10^6^/ejaculate, total sperm motility ≥40% and normal sperm morphology ≥ 4% [50]) were obtained from seven men over the age of 40 years either undergoing a routine semen analysis or ART procedure at Repromed, a private IVF clinic, in 2018. Men diagnosed with oligo- or azoospermia, or who had a known infectious status, were excluded from the study. Samples were de-identified before they were made available to researchers. Ethical approval was obtained from the University of Adelaide Human Research Ethics Committee (HREC approval number: H-2017-021), as per the Australian National Health and Medical Research Council (NHMRC) Ethical Guidelines. Patient consent was waived by the HREC as involvement in the research carried no more than low risk. Repromeds Scientific Advisory Board also approved the study.

### 2.2. Animals and Ethics Approval

In this study, 12–18-month-old male studs C57BL6 × CBA (CBAF1) to mimic advanced paternal age [51], pre-pubertal (3–4-week-old) CBAF1 female mice as oocyte donors, 6-month-old vasectomized males CBAF1 to induce pseudo pregnancy and Swiss female mice (7–9-week-old) as recipients for embryo transfer were used. Animals were housed in a temperature-controlled holding room at 21 °C, on a 12:12 h light/dark cycle. All mice were fed standard chow and acidified water ad libitum. The use and care of all animals used in the study was approved by the Animal Ethics Committee of The University of Adelaide (M-2013-119) under the guidance of the Australian Code for the Care and Use of Animals for Scientific Purposes, 8th edition 2013.

### 2.3. Culture Media

For the treatment of sperm with idebenone (Sigma-Aldrich, Castle Hill, New South Wales, Australia), G-IVF PLUS medium (Vitrolife, Denver, CO, USA) was supplemented with either 5 µM or 50 µM of idebenone, with these concentrations previously been shown to reduce ROS related cellular damage up to 50% in brain cells and prevent ROS related cell death in retinal epithelium [46,48,52,53]. G-IVF PLUS medium was used as a control. For embryo culture, G1.3 PLUS and G2.3 PLUS (Vitrolife) was used. G-IVF PLUS and G1/G2 media were equilibrated at 37 °C 6% CO_2_, 5% O_2_, 89% N_2_ for ≥ 4 h prior to use. G-MOPS PLUS handling medium (Vitrolife) was warmed to 37 °C and used for the transport handling of mouse oocytes, epididymis and embryos. A within-subjects design was used for the assessment of all experiments, such that sperm from one human/mouse was represented across all treatment groups.

### 2.4. Isolation of Human Motile Sperm

To isolate motile sperm fractions that would normally be used for ART insemination, human liquefied semen samples within 1 h of ejaculation were washed of seminal plasma by a swim-up technique [50]. Whole semen was split into three aliquots and underlaid under 2 mL of G-IVF PLUS media (control: 0 µM) or G-IVF PLUS medium supplemented with either 5 µM or 50 µM of idebenone and incubated for 1 h at 37 °C 6% CO_2_, 5% O_2_, 89% N_2_. This is the standard length of time for a swim-up in clinical IVF [50]_._ Progressively motile sperm were isolated from the top 400 µL of medium. All swim-ups contained ≥90% progressively motile sperm and a concentration of ≥1 × 10^6^/mL, which were assessed on a Makler (SEFI-Medical, Haifa, Israel) under 20× phase contrast light microscopy.

### 2.5. Mouse Sperm Collection

Following euthanasia, sperm from the cauda epididymis and vas deferens were extracted and incubated in G-IVF PLUS (Vitrolife) control medium or G-IVF PLUS medium supplemented with 5 µM, or 50 µM idebenone and incubated for 1 h at 37 °C 6% CO_2_, 5% O_2_, 89% N_2_.

### 2.6. Sperm Intracellular (DCFDA) and Superoxide ROS (MSR) Assessment

Sperm (1 × 10^6^/mL) were assessed for intracellular ROS concentration by incubation with 5 µM DCFDA (2′,7′-dichlorodihydrofluorescein diacetate, H_2_DCFDA; Life Technologies, Carlsbad, CA, USA) for 60 min at 37 °C 6% CO_2_, 5% O_2_, 89% N_2_ and counterstained with 1 µM Propidium Iodide (PI) (Sigma-Aldrich) for 5 min [54]. For assessment of superoxide ROS, sperm were incubated in 5 µM MitoSOX Red (MSR, Molecular Probes, Eugene) and 5 µM SYTOX Green live/dead counterstain for 30 min at 37 °C 6% CO_2_, 5% O_2_, 89% N_2_ [55]. Following incubation, samples were washed and suspended in phosphate-buffered saline (PBS) (Sigma-Aldrich) before assessment on a BD FACs Canto II Flow Cytometer (BD Bioscience, North Ryde, Australia), which had CST beads run daily to ensure fluorescence was kept consistent on measurement days. Then, 10,000 cells were examined per sample and non-specific events, including cells identified as positive for PI or SYTOX Green, were gated out. Negative controls were included where sperm were only incubated in either PI or SYTOX Green respectively. The concentrations of ROS for each probe were expressed as the mean fluorescence intensity (fluorescence units) for live cells.

### 2.7. Mouse Oocyte Collection and IVF

Pre-pubertal (3–4-week-old) CBAF1 females were super-ovulated by intra-peritoneal injection of 5IU of Pregnant Mare’s Serum Gonadotropin (PMSG; Folligon, Intervet, Bendigo, Australia), followed by injection of 5IU of human chorionic gonadotropin (hCG; Pregnyl; Organon, Oss, The Netherlands) 48 h later. At 12.5 h post hCG injection, cumulus-enclosed oocytes were collected and incubated in 80 µL drops of G-IVF PLUS medium (Vitrolife), pre-equilibrated at 37 °C: 6% CO_2_; 5% O_2_; 89% N_2_ under an oil overlay (Ovoil, Vitrolife), before insemination with 1 × 10^5^/mL sperm from either control or 5 µM idebenone treatment. Prior to insemination, sperm motility was assessed under phase contrast light microscopy classifying 200 spermatozoa as either progressive motile, non-progressively motile or immotile. Gametes were co-incubated for 4 h at 37 °C, 6% CO_2_; 5% O_2_; 89 % N_2_. Following co-incubation, putative zygotes were removed and cultured in 20 µL drops of pre-equilibrated G1+ culture medium (Vitrolife), under an oil overlay, for 48 h at 37 °C: 6% CO_2_; 5% O_2_; 89% N_2_. Fertilization was assessed at 24 ± 1.5 h post-insemination and was identified by cleavage to the 2-cell stage. Embryos were scored for 8-cell development at 48 ± 1.5 h post insemination (Day 3) and at this stage were transferred to 20 µL drops of pre-equilibrated G2 PLUS medium and cultured at 37 °C: 6% CO_2_; 5% O_2_; 89% N_2_ for a further 48 h. At 96 ± 1.5 h post-insemination (day 5) blastocyst development was assessed. Blastocysts were categorized as either an early blastocyst (blastocoel cavity <2/3 of embryo), blastocyst, expanded blastocyst (blastocoel cavity >2/3 of embryo) or hatching blastocyst (protrusion of the blastocyst through the zona pellucida) [56]. Embryos were then fixed in 4% paraformaldehyde in PBS (Sigma-Aldrich) or vitrified.

### 2.8. Blastocyst Cell Differentiation and Cell Apoptosis Assessments

Day 5 blastocysts were fixed overnight in 4% paraformaldehyde (Sigma-Aldrich) at 4 °C and transferred to 8% polyvinylpyrrolidone in phosphate-buffered saline (PBS/PVP) the following day. To assess cell differentiation and cellular apoptosis, fixed blastocysts were incubated in 0.1 M glycine in PBS/PVP for 5 min before they were permeabilized in 0.25% Triton X-100 (TX) in PBS/PVP (Sigma-Aldrich) for 15 min at room temperature (RT). Blocking for non-specific antibody binding was achieved by overnight incubation in 10% Donkey serum in PBS/PVP at 4 °C. The following day, blastocysts were incubated in anti-Oct-4 primary antibody (1:100 dilution in TX; Santa Cruz Biotechnology, Dallas, TX, USA) to assess cell differentiation or anti-active Caspase-3 primary antibody (1:100 dilution in TX; Abcam, Cambridge, UK) for cell apoptosis assessments, for 1.5 h at 37 °C. This was followed by incubation with Alexa-Fluor donkey anti-goat secondary antibody (1:100 dilution in PBS/PVP; Life Technologies) or Alexa-Fluor 488 donkey anti-rabbit antibody (1:100 dilution in PBS/PVP; Life Technologies) for 2 h at RT. Blastocysts were counterstained with Hoechst nuclear stain (1:400 dilution in PBS/PVP; Life Technologies) for 5 min at RT before they were individually mounted onto glass slides using glycerol, and visualized using an Olympus BX51 Epifluorescence Microscope (Olympus America, Center Valley, PA, USA), at 20× magnification and an excitation of 350–510 nm. Cells which were positive for Oct-4 (red) or active Caspase-3 (green) and Hoechst (blue) stains were counted to obtain inner cell mass, apoptotic cells and total cell number, respectively. Trophectoderm cell number was determined by subtracting the number of Oct-4 positive cells from the number of Hoechst-positive cells. For cells to be classified as apoptotic, Caspase-3 (green) had to co-localize with nuclear (Hoechst–blue) staining. Images were taken using SPOT Advanced Software and assessed using Fiji ImageJ [57].

### 2.9. Embryo Vitrification and Uterine Transfer

Following 96 h of culture (day 5), blastocysts were vitrified using the RapidVit-Blast media system (Vitrolife) in groups of seven or eight, as per the manufacturer’s instructions, and stored in liquid nitrogen. The warming of embryos was conducted on the morning of embryo transfer using the Warm-Cleave Media System (Vitrolife). The inserts of the Rapid-I Straws containing the embryos were rapidly submerged into Warm1-Cleave medium for 30 s followed by the transfer of embryos to Warm2-Cleave, Warm3-Cleave and G-MOPS PLUS handling medium for 1 min, 2 min and 3–5 min, respectively. The embryos were then placed into pre-equilibrated G2 PLUS medium under an oil overlay (Ovoil, Vitrolife) for at least 4 h at 37 °C, 6% CO_2_, 5% O_2_, 89% N_2_ prior to embryo transfer to allow for re-expansion. Cryosurvival rates were assessed prior to embryo transfer. Swiss female mice were mated with vasectomized CBAF1 males, and the presence of a copulatory plug was determined on day 1 of pseudopregnancy. At 2.5 days of pseudo-pregnancy, females were anaesthetized with 2% Avertin (Sigma-Aldrich), before six thawed blastocysts (≥expanded blastocyst) from either the control or 5 µM idebenone were transferred to contralateral uterine horns of 10 mothers (total of 60 blastocyst per treatment group), meaning that embryos from both treatment groups were gestating in the same mother. On day 18 of gestation, the total number of pregnancies, implantation sites, resorptions and viable fetuses were examined. Fetal and placental lengths/diameters, and weights were measured [17].

### 2.10. Statistics

Statistical analysis was performed in SPSS (SPSS Version 18, SPSS Inc., Chicago, IL, USA) or GraphPad Prism (GraphPad Software v6, San Diego, CA, USA). Sperm ROS concentrations, motility and embryo development were analyzed by either a repeated measure ANOVA with a Tukey’s multiple comparison test for multiple groups or a paired t-test when comparing only two groups. Embryo cell numbers, apoptosis and blastocyst cryosurvival rates were assessed by a General Linear Model with both treatment group and male fitted as fixed effects and staining/warming day replicate fitted as a covariate. For embryo transfer data, fetal outcomes were analyzed by a univariate GLM, with mother included as a random effect to control for variations to fetal outcomes induced by mother. There was no effect of date of transfer, uterine horn or litter size, and therefore, these were not added to the final model. A *p* < 0.05 was considered significant with a *p* < 0.1 classified as a trend.

## 3. Results

### 3.1. Idebenone and Sperm Intracellular and Superoxide ROS Concentrations in Advanced Paernal Age

Seven normospermic men with a mean age of 43 years (range: 41–46 years) and a mean body mass index of 27.5 kg/m^2^ (range: 24.4–30.7 kg/m^2^) were assessed for both intracellular and superoxide sperm ROS concentrations following incubation in either 0 µM (control), 5 µM or 50 µM of idebenone for 1 h during sperm washing. Although, there was no effect of idebenone on sperm intracellular ROS concentrations (Figure 1a,b, *p* > 0.05), 5 µM of idebenone reduced sperm superoxide concentrations compared with control (*p* < 0.001, Figure 1c,d) and 50 µM of idebenone (*p* < 0.01, Figure 1c,d). Of note, 50 µM of idebenone had no effect on sperm superoxide concentrations compared with control (*p* > 0.05, Figure 1c,d).

We then wanted to confirm our findings from humans in our advanced paternal age mouse model. In aged male mice, the addition of both 5 µM and 50 µM idebenone for 1 h reduced sperm intracellular ROS concentrations compared with control (*p* < 0.05, Figure 1e,f). Similar to what was seen in the human, only 5 µM of idebenone significantly reduced sperm superoxide concentrations compared with the control (*p* < 0.01, Figure 1g,h), although no differences were seen between 5 µM and 50 µM idebenone (*p* > 0.05, Figure 1g,h). Due to the results seen in humans and mice, we continued to only assess the impact of 5 µM of idebenone on sperm for defining subsequent effects on fertilization, embryo development and implantation rates.

### 3.2. Idebenone and Fertilization Rates and Embryo Development in a Mouse Model of Advanced Paternal Age

The addition of 5 µM of idebenone to sperm prior to IVF insemination increased the number of progressively motile sperm compared with the control (*p* < 0.01, Figure 2A), which led to an increase in fertilization rates (*p* < 0.05, Figure 2B). Sperm superoxide concentrations independent of idebenone treatment were moderately negatively correlated with fertilization rates (−0.370, *p* = 0.06), such that higher sperm superoxide concentrations were associated with a lower fertilization rate. There was no effect of 5 µM idebenone on subsequent day 3 on-time development (% 8-cells) (*p* > 0.05, Figure 2C) or day 5 blastocyst development compared with control (*p* > 0.05, Figure 2D,E).

### 3.3. Idebenione and Blastocyst Differentiation and Appoptosis in a Mouse Model of Advanced Paternal Age

We assessed embryo quality markers of blastocyst cellular differentiation (Oct4 staining) and apoptosis (measured by Caspase 3); 5 µM of idebenone to sperm prior to IVF insemination increased blastocyst total cell number (*p* < 0.001, Figure 2F,J), trophectoderm cell number (*p* < 0.01, Figure 2G) and inner cell mass cell number (*p* < 0.01, Figure 2H) compared with the control. We saw no change in the percentage of cells positive for Caspase 3 in the blastocysts due to incubation of sperm with 5 µM of idebenone prior to IVF (*p* > 0.05, Figure 2I,K), indicating that the increase in cell numbers was not due to a reduction in cellular apoptosis.

### 3.4. Idebenone and Blastocyst Thaw Survival Rates and Embryo Implantation Rates in a Mouse Model of Advanced Paternal Age

Given the positive benefits of 5 µM idebenone to sperm prior to IVF on embryo cell numbers, we then wanted to assess if this would translate to improved implantation and fetal development following embryo transfer. To maximize our embryo transfer protocol and to better mimic procedures that match the ART clinic, we performed blastocyst vitrification and subsequent warming prior to embryo transfer. Note that 5 µM of idebenone to sperm prior to IVF insemination increased blastocyst survival rates following embryo warming compared with controls (*p* < 0.01, Figure 3a). Although, we saw no difference in the number of mothers that became pregnant between the two groups (*p* > 0.05, Figure 3b), 5 µM of idebenone to sperm prior to IVF increased the proportion of blastocyst that implanted compared with controls (*p* < 0.01, Figure 3c). There was no effect of 5 µM of idebenone to sperm prior to IVF on the number of fetuses per implantation, fetal length or fetal weights (*p* > 0.05, Figure 3d–f). Interestingly, 5 µM of idebenone to sperm prior to IVF insemination reduced placental area (*p* = 0.07, Figure 3g) and placental weights (*p* < 0.05, Figure 3h) compared with the control, resulting in an increased fetal-to-placental weight ratio (*p* = 0.07, Figure 3i).

## 4. Discussions

While originally it was perceived that male aging had no influence on fertility, we now know that advanced paternal age is associated with subfertility, manifesting as increased time to conception, reduced implantation rates, increased miscarriage rates and reduced live birth rates in both general and ART populations [58,59]. A contributing factor to these poorer fertility outcomes in older men is increased sperm ROS concentrations and associated DNA damage [11,12,13,14], which, independently, are linked to reduced pregnancy rates and increased miscarriage rates [60,61]. Although, unlike other biological and lifestyle causes of sub fertility, there are limited intervention options that can be used to reverse the effects of biological aging. One potential way to combat increased sperm ROS concentrations associated with advanced paternal age could be through the addition of antioxidants both orally or to sperm culture media prior to IVF. In this proof-of-concept study, we have shown that the addition of idebenone to sperm culture media prior to IVF insemination is able to reduce sperm superoxide ROS concentrations associated with advanced paternal age. This was then associated with an increase in fertilization rates following IVF insemination in mice, increased blastocyst cell numbers, improved blastocyst survival rates following embryo warming and increased implantation rates following embryo transfer, all of which are indicative of better overall embryo quality.

Idebenone was used, due to its chemical similarities to co-enzyme Q10 (a potent antioxidant) as well as its mitochondrial permeability (main source of ROS production in sperm [25]), and its water-soluble properties (making it suitable for use in gamete culture media) [43]. When added to sperm washing media, idebenone was able to reduce sperm ROS concentrations in both males >40 years and in our mouse model of advanced paternal age, although there were some slight differences seen between species. For instance, both 5 µM and 50 µM of idebenone added to mouse sperm were able to reduce hydrogen peroxide and hydroxyl radicals (DCFDA); however, the same result was not seen in humans, while superoxide production in both species was reduced with the addition of 5 µM of idebenone. Given that idebenone is labeled as a mitochondrial permeable antioxidant [62], is involved in redox cycling [45] and can scavenge a variety of free radical species with concentrations above ~2 µM [45], it was expected that it would be able to reduce superoxide anion production in sperm at 5 µM. Further, idebenone is also known to have hydrogen peroxide scavenging properties [63,64], which could also explain why it was able to reduce hydrogen peroxide and hydroxyl radicals in the mouse. We believe that the likely reason why mouse sperm were able to reduce both the hydrogen peroxide and hydroxyl radicals, while the same was not seen in humans, was due to the reduced biological and lifestyle variability of research rodents. The mice used, while outbred, were kept in a very controlled research environment limiting other variables (i.e., nutrition) that could modify baseline ROS concentrations, resulting in more consistent results. Although men in this cohort were of an advanced age (>40 years), other biological and lifestyle factors (not obtained) could have resulted in some men having higher baseline ROS concentrations, which may require a higher concentration (>50 µM) of idebenone to illicit a reduction in hydrogen peroxide and hydroxyl radicals.

The addition of 5 µM idebenone to sperm culture media prior to IVF was associated with improved fertilization rates in our aged mice, with lower fertilizations rates following standard insemination (IVF) a common feature seen in advanced paternal age [58,59,65]. Increased sperm superoxide production coincides with increased formation of lipid aldehyde 4-hydroxynonenal (4HNE) in sperm [66] which is also seen in aging [67]. The formation of 4HNE can impair the fertilizing capacity of sperm by oxidizing varies surface proteins (i.e., HSPA2) involved in sperm oocyte fusion [68]. This results in a decrease in the number of sperm that can bind to the zona pellucida [17,69,70] and subsequently reduces fertilization. Another way that idebenone may be contributing to the improvements seen in fertilization rates in our mouse model of advanced paternal age is from the benefits seen to sperm progressive motility. In addition to its scavenging antioxidant properties, idebenone also acts as an electron donor to detoxify radicals as well as an aide in ATP production once reduced to its hydroquinone form, idebenol, by NAD(P)H quinone oxidoreductase 1 (NQO1) [71]. Sperm contain NAD(P)H quinone oxidoreductase (NQO also known as DT- diaphorase [72,73]) and a sperm-specific enzyme with diaphorase activity [74], suggesting that bioactivation of idebebone can occur within sperm. Idebenone can increase/restore ATP levels under stress conditions with impairments to mitochondrial complex 1 function, by shuttling electrons to complex III of the electron transport chain [63,75]. Mitochondrial energy metabolism is crucial for male reproductive function, with aging associated with reduced mitochondrial respiration rates of sperm [76,77,78] and decreased sperm motility [9]. Thus, idebenone may be increasing the motility and fertilizing capacity of sperm from older males, potentially via improvements to sperm mitochondrial function and ATP production; however, furthers studies are required to test this.

We were able to also show that the addition of 5 µM idebenone to sperm culture media prior to IVF was associated with improve embryo quality as measured by blastocyst cell numbers, blastocyst cryosurvival rates and increased implantation rates following embryo transfer. Blastocyst cell numbers, or the distribution of cells (trophectoderm vs. inner cell mass) within the blastocyst, positively correlates with implantation and pregnancy rates as well as fetal growth [79,80], while cryosurvival of blastocysts is also associated with better embryo quality as only those blastocysts of high quality are able to survive the considerable morpho-functional damage that occurs during cryopreservation and warming [81]. It has already been previously shown that embryos generated (in mice and cattle) from sperm with high ROS concentrations have reduced blastocyst quality (reduced cell numbers and implantation rates) [18,82]. Therefore, the reductions to sperm ROS concentrations due to idebenone could be one possible reason for the improved embryo quality seen in sperm from our aged males. Recently, idebenone has been shown to have additional molecular activity outside that of oxidative damage protection. These include [44]: improved mitochondrial respiration (as discussed above), selective PPARα/γ agonist, inhibition of p52Shc which acts as an adaptor protein required for a variety of molecular complexes, most notability protein tyrosine kinase receptors (which are vital in sperm capacitation, acrosomal exocytosis and gamete fusion [83]), upregulation of *Lin28A* (although unlikely to be of importance to sperm, due to lack of transcription), reductions to inflammation (commonly seen in aging [84]) and reductions to endoplasmic reticulum stress (again commonly seen in aging [85]). Therefore, improvements to some of these other molecular pathways of aged sperm by idebenone could also be contributing to the associated benefits seen on downstream embryo development and health.

Sperm DNA damage, a major manifestation of oxidative stress and elevated ROS [86,87,88,89,90], is associated with delayed blastocyst development and decreased pregnancy rates [91]. Caspase-3 is a protein which plays a fundamental role in cellular apoptosis, and may be activated via intrinsic or extrinsic apoptotic pathways [92]. It is present in mouse pre-implantation embryos [93], and its activation may be a downstream effect of decreased mitochondrial function and DNA damage, associated with elevated ROS [92,93]. Additionally, idebenone has been shown to suppress levels of active Caspase-3 in human umbilical cord vascular endothelial cells [47] and rat livers [94], and therefore was selected for use as a marker of apoptosis in this study. Whilst the relationship between sperm oxidative stress and apoptosis in the blastocyst had been previously shown [33], our study did not show a reduction in active Caspase-3 positive cells in blastocysts produced from aged sperm with or without 5 µM idebenone. A limitation of measuring active Caspase-3 in blastocysts is that it may not be reflective of the sperm DNA damage inherited by the embryo, which may be better detected by gammaH2AX staining or the comet assay. However, given that blastocyst cell numbers were increased in embryos produced from idebenone-treated sperm, it suggests that this increase in cell numbers was more likely due to an increase in cellular division during embryogenesis and not due to a reduction in cellular apoptosis. Further, there is evidence that the quality of the oocyte impacts the ability for it to repair sperm DNA damage at fertilization, with younger oocytes (as seen in our model) able to repair DNA damage at a better rate than aged oocytes [95]. As advanced paternal age tends to coincide with advanced maternal age, outcomes may be different if we had utilized aged oocytes in our IVF model.

A higher paternal age at conception has been previously shown to be associated with lower infant birthweight and increased risk of premature birth [96]. In our study, while we saw no effect of 5 µM idebenone to sperm prior to IVF on fetal weights and lengths, we did find that idebenone decreased placental weights leading to a trend for increased fetal to placental weight ratios. Fetal-to-placental weight ratios are an important indicator of nutrient transfer from the placenta to the fetus and thus can be used as a surrogate measure of placental efficiency [97], with an increase in fetal-to-placental weight ratio as seen in idebenone-treated sperm associated with upregulated placental nutrient transfer capacity [97]. Interestingly, advanced paternal age at conception is also associated with increased placental weights in humans [96] with high ROS concentrations in sperm also associated with higher placental weights and a decreased fetal-to-placental weight ratios in a mouse model [18]. Therefore, placental modifications due to advanced paternal age at conception could be directly related to increased sperm ROS concentrations, with antioxidants aimed at reducing sperm ROS concentrations prior to fertilization able to improve placental function. This is important, as placental function in utero is directly related to chronic disease risk in adulthood and will be the focus of further studies [98].

This study has also highlighted an important factor that may be contributing to the limited clinical uptake of antioxidants in human sperm culture media, which is that excessive amounts of antioxidants can interfere with physiological ROS concentrations, leading to enhanced ROS generation in mitochondria and further oxidative injury to cells [99,100]. In this study, 50 µM of idebenone had no effect on sperm superoxide levels in both human and mouse. Previous studies have shown that idebenone can act as a pro-oxidant through the inhibition of complex 1 of the electron transport chain and an electron acceptor, thereby promoting superoxide production [101,102,103], with concentrations >25 µM causing apoptosis in human neuronal cell lines in vitro [65]. Therefore, its pro-oxidant properties coupled with a higher concentration (i.e., 50 µM) may have the opposite effect, leading to a state of reductive stress and no change to sperm ROS concentrations [99,100]. It is, however, important to note that sperm require a physiologically balanced level of ROS in order to function [15,16,66] and that the dose of antioxidants added to culture media must be carefully considered as they may not be suitable for men with “normal” ROS concentrations and, if used incorrectly, could in fact result in an inhibitory effect [66]. Further, in this study, we only assessed one time point (1 h) of antioxidant exposure; however, in clinical IVF, sperm can sit in media for up to five hours prior to insemination. Understandingly, antioxidant exposure lengths on sperm quality and function will be important for understanding its true clinical utility.

## 5. Conclusions

This proof-of-concept study has shown that the addition of a cell permeable antioxidant is able to reduce sperm ROS concentrations and is associated with improved subsequent embryo quality in models of advanced paternal age. Further studies to determine the optimal doses, best antioxidant types, exposure length to sperm in vitro, understanding those patients who would benefit the most and downstream consequences to embryo quality and pregnancy outcomes in humans are still required to fully understand whether or not it is plausible and clinically safe to add antioxidants to sperm preparation media.

## Figures and Tables

**Figure 1 antioxidants-10-01079-f001:**
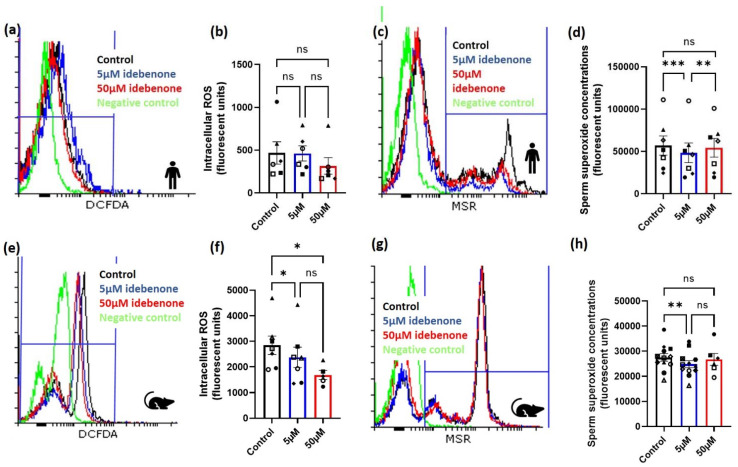
Idebenone in vitro reduces sperm reactive oxygen species concentrations in advanced paternal age. (**a**) Representative FACS histograms for human sperm for intracellular ROS, (**b**) Human sperm intracellular ROS concentrations following 1 h incubation in differing concentrations of idebenone, (**c**) Representative FACS histograms for human sperm for superoxide, (**d**) Human sperm superoxide concentrations following 1 h incubation in differing concentrations of idebenone, (**e**) Representative FACS histograms for mouse sperm for intracellular ROS, (**f**) Mouse sperm intracellular ROS concentrations following 1 h incubation in differing concentrations of idebenone, (**g**) Representative FACS histograms for mouse sperm for superoxide, (**h**) Mouse sperm superoxide concentrations following 1 h incubation in differing concentrations of idebenone. N = 6 human sperm biological replications for intracellular ROS and N = 7 human sperm biological replicates for superoxide concentrations. N = 6 mouse sperm biological replicates for intracellular ROS and N = 11 mouse sperm biological replicates for superoxide concentrations. Data were analyzed by a repeated measure ANOVA with a Tukey’s multiple comparison test. Different symbol shapes represent the same biological sample; 5 µM—5 µM idebenone, 50 µM—50 µM idebenone, ns—not significant. * *p* < 0.05, ** *p* < 0.01, *** *p* < 0.001.

**Figure 2 antioxidants-10-01079-f002:**
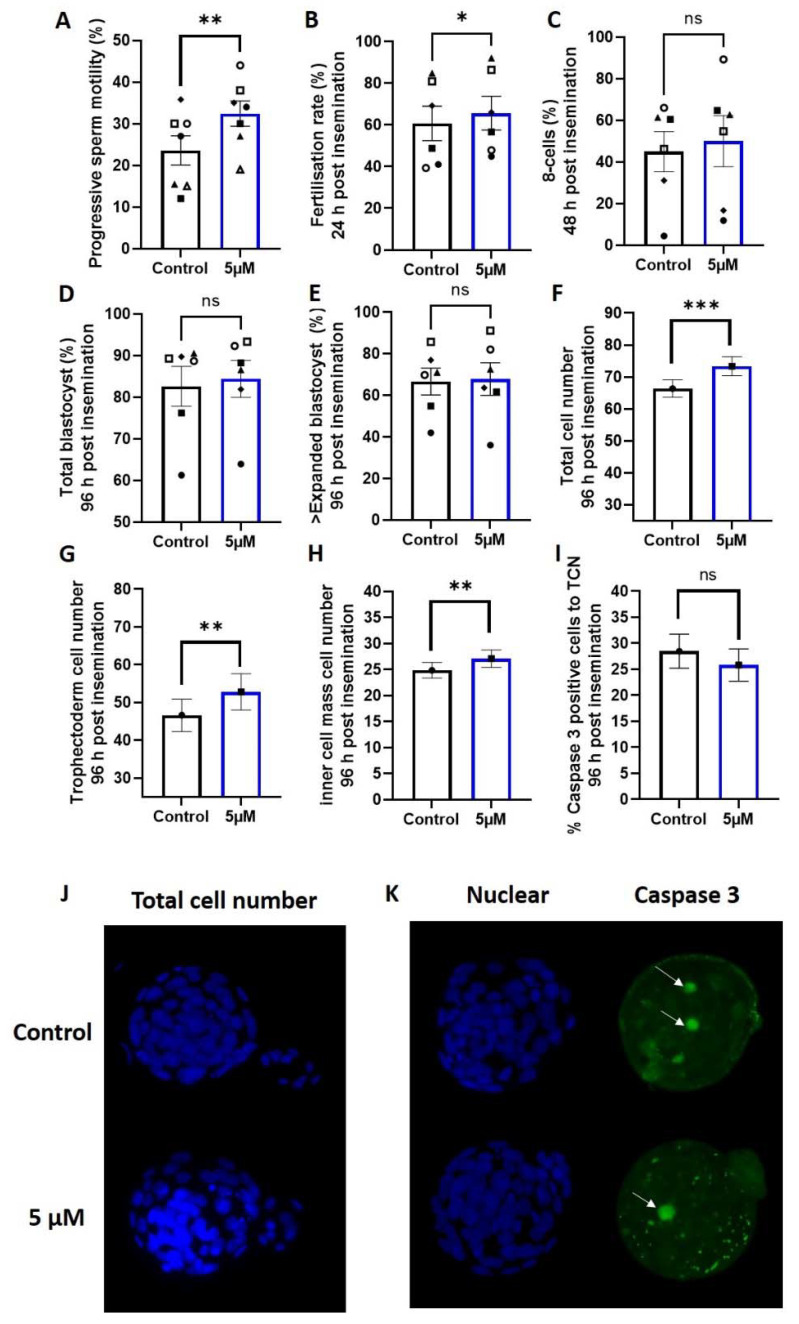
5 µM idebenone in vitro increases sperm motility, fertilization rates and total cell numbers in embryos generated from advanced paternal age. (**A**) Progressive sperm motility, (**B**) Fertilization rate from number of MII oocytes inseminated, (**C**) Percentage of 8 cell embryos 48 h post insemination from number that fertilized, (**D**) Total blastocyst (≥early blastocyst) 96 h post insemination from number that fertilized, (**E**) Percentage of ≥ expanded blastocyst (on time) 96 h post insemination from number that fertilized, (**F**) Total cell number of blastocyst 96 h post insemination, (**G**) Trophectoderm cell number of blastocyst 96 h post insemination, (**H**) Inner cell mass cell number of blastocysts 96 h post insemination, (**I**) Percentage of caspase 3 positive cells in blastocysts 96 h post insemination, (**J**) Representative images of total cell number (blue) for control and 5 µM idebenone generated blastocyst 96 h post insemination and (**K**) Representative images of Caspase 3 positive staining (green) for control and 5 µM idebenone generated blastocysts 96 h post insemination. 804 control and 788 5 µM idebenone embryos assessed for embryo morphology from 6 male biological replicates; 32 control and 27 blastocyst assessed for blastocyst differentiation from 3 male biological replicates and 44 control and 51 5 µM idebenone blastocyst assessed for Caspase 3 from 4 male biological replicates. Different symbol shapes represent the same biological sample. White arrows point to Caspase 3 positive cells. Sperm motility and embryo development were assessed by a paired T-test, while embryo cell numbers and apoptosis were assessed by a General Linear Model with both treatment group and male fitted as fixed effects and staining replicated fitted as a covariate. TCN—Total cell number, 5 µM—5 µM idebenone, ns—not significant. * *p* < 0.05, ** *p* < 0.01, *** *p* < 0.001.

**Figure 3 antioxidants-10-01079-f003:**
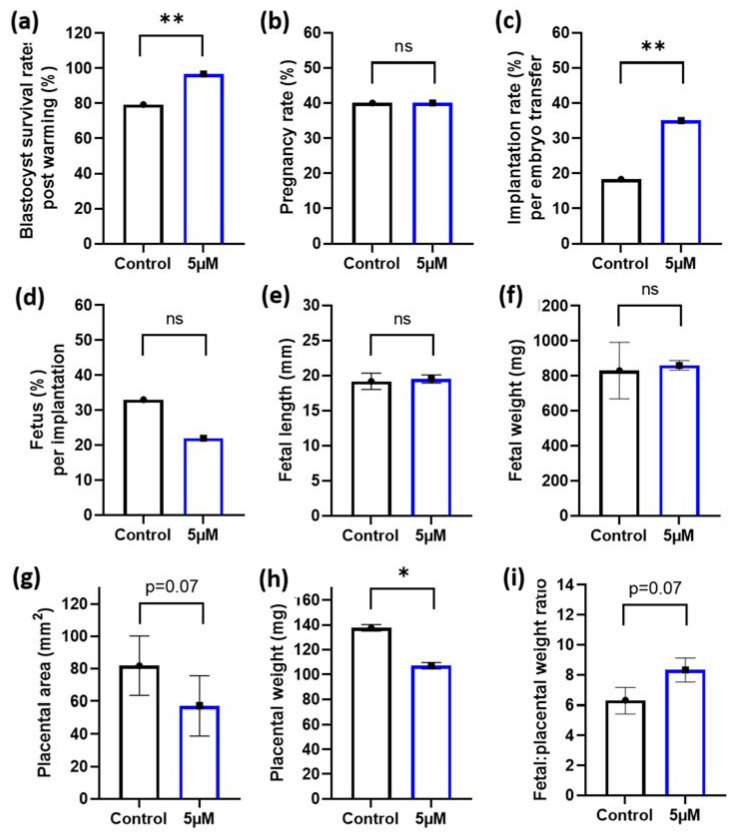
5 µM idebenone in vitro to sperm prior to IVF increases blastocyst thaw survival rates and implantation rates following embryo transfer. (**a**) Percentage of blastocyst that survived embryo thawing post embryo vitrification, (**b**) The percentage of pregnant mothers per embryo transfer, (**c**) The percentage of both resorptions and fetus per embryo transferred, (**d**) The percentage of fetus per implantation site, (**e**) Fetal lengths, (**f**) Fetal weights, (**g**) Placental area, (**h**) Placental weight and (**i**) Fetal placental weight ratios. Blastocyst thaw survival rates represents 92 blastocysts per treatment group. Embryo transfers and pregnancy data is representative of 60 blastocyst per treatment group from 6 male biological replicates transferred into 10 recipient mothers (6 blastocyst per treatment group per uterine horn). Blastocyst cryosurvivals were analyzed by a General Linear Model (GLM), with both treatment group and male fitted as fixed effects and warming day fitted as a covariate. Embryo transfer data and fetal outcomes were analyzed by a univariate GLM, with mother included as a random effect to control for variations to fetal outcomes induced by the mother: 5 µM—5 µM idebenone, ns—not significant. * *p* < 0.05, ** *p* < 0.01.

## Data Availability

Data availability can be accessed by contacting the corresponding author.

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
