# Peer review of "Improving Sperm Oxidative Stress and Embryo Quality in Advanced Paternal Age Using Idebenone In Vitro—A Proof-of-Concept Study"

_antioxidants, 2021, doi:10.3390/antiox10071079_

Round 1
Reviewer 1 Report
The manuscript by Nikitaras et al represents an interesting proof of concept study that aimed to demonstrate that idebenone can improve embryo quality and reduce oxidative stress levels in sperm. The manuscript extends a long line of papers that have tested various antioxidants for this purpose. While the experimental details are straight forward, there are several theoretical, conceptual, and experimental issues with this manuscript. As such, the experimental design was geared towards finding an antioxidative effect but fell short by the specific methods used, which is based on an incomplete understanding what idebenone can do and how it is believed to do that.
Major points
- If advanced paternal age, which is even mentioned in the title was a key issue of the study, why was there not a cohort of younger men added to the study (in addition to the older men included) to demonstrate age dependency? Can we even expect to see a reduction of ROS in this cohort? It might be better to remove this part of the study without an adequate control.
- Conceptionally, there is an issue to portrait idebenone as an antioxidant only (see current literature). Especially given the very small effect on oxidative stress levels in sperm.
- It is well known and described in reference 41 that idebenone, to act as antioxidant, has to be bioactivated (reduced). Do the authors know about the presence of a bioactivating system in sperm? If so, why was this not shown/described/cited? If not, the study approach lacks a major justification. The cited references used isolated mitochondria supplied with substrates to fuel a complex II dependent bioactivation. Is is not what happens in intact cells.
- The bioactivation of idebenone also reduces the latent pro-oxidative activity of idebenone, this feature was not discussed although it could explain the lack of dose effect observed in Figure 1 that the authors address in the discussion.
- It seems that the authors conceptually used idebenone as a simple antioxidant and did not even contemplate the notion that the observed effects for example on blastocyst survival and implantation rate could be caused by large number of additional effects unrelated to antioxidant effects. See for example Lei et al. 2018
- If idebenone is supposed to be portrayed as a “better” antioxidant (line 90), why was no control group included to show its superiority?
- The antioxidant effect of idebenone was previously predominantly shown as a reduction of lipid peroxidation. It is therefore unexplained why this endpoint was not assessed (by using for example BODIPY-C11 dye) but instead water-soluble ROS dyes such as DCFDA were used. Could this be the reason why no major antioxidant effects were observed? As an alternative to measuring ROS, measuring ROS-induced damage (i.e. lipid peroxidation, protein-nitrosylation or oxidative DNA damage via 8-oxo-dG) could have been more informative.
- Protection against pro-oxidative damage (i.e. DNA damage) could have supported the presence of an antioxidant effect. However, it is unclear why DNA damage was not directly assessed. In contrast caspase 3 staining was employed. The caspase staining in Figure 2k is not convincing, it even shows a higher signal with idebenone (it is unclear what we are looking at in this image, what are the bright spots and what exactly was quantified, how about a counterstain with DAPI to get an orientation?). DNA damage could have been much more directly and convincingly demonstrated using comet assays or gammaH2AX staining. It could have also answered the question if the DNA damage in the blastocysts was inherited from the sperm or originated in the blastocysts.
- The discussion on the properties of idebenone is too simplistic at several levels: for example the interaction with complex III of the mitochondria (line 398) was only ever demonstrated when complex I was impaired. Is there any evidence that this happens in aged sperm? What about all the other effects idebenone has on cells?
- The study does not provide any evidence of a causative nature that antioxidant function (if indeed present) is responsible for improved embryo quality. Therefore, it is descriptive and as such should be discussed in this manner.
- Given the extent of literature on antioxidants (or compounds presumed to be antioxidants) and sperm quality, I do not think that the concluding statement is adequate and should be reworded.
Minor points:
- How were ROS levels quantified in Figure 1? Which peak was used for quantification?
- The references for the antioxidant effect of idebenone (42,47) are really old and should be replaced with contemporary references that are suited to support the statements. Especially since both references refer to cell free studies and are therefore unsuited to justify the doses used with intact cells. Again, the bioactivation process must be accounted for.
- The figure legends do not illustrate, which statistical tests were used for each figure. This makes the interpretation of the validity of the results impossible. For example, I struggle to see how Figure 2b can show a statistically significant effect based on the individual samples shown as dots. Illustration of how the samples from each donor reacted by connecting the relevant sample dots with lines might illustrate this effect better.
- Line 91: vitamins E and C are cell permeable and especially vitamin C as an acid is very water soluble. Therefore, the statement that they are ALL lipophilic is plain wrong. While I fully agree that CoQ10 is unsuited for the purpose of this study as in vitro it takes at least 9 days to reach the mitochondria, vitamin C and E will reach them in a few minutes.
- To how much idebenone were the oocytes possibly exposed to during the procedure?
Author Response
Reviewer 1
The manuscript by Nikitaras et al represents an interesting proof of concept study that aimed to demonstrate that idebenone can improve embryo quality and reduce oxidative stress levels in sperm. The manuscript extends a long line of papers that have tested various antioxidants for this purpose. While the experimental details are straight forward, there are several theoretical, conceptual, and experimental issues with this manuscript. As such, the experimental design was geared towards finding an antioxidative effect but fell short by the specific methods used, which is based on an incomplete understanding what idebenone can do and how it is believed to do that.
Major points
Comment 1: If advanced paternal age, which is even mentioned in the title was a key issue of the study, why was there not a cohort of younger men added to the study (in addition to the older men included) to demonstrate age dependency? Can we even expect to see a reduction of ROS in this cohort? It might be better to remove this part of the study without an adequate control.
Response: The purpose of this proof of concept study was to see if we could reduce ROS production in sperm from older men, a hallmark phenotype of aged sperm (PMID: 18342194, 33660452), not to prove that older men have increased ROS production or that we could reduce levels back to that of younger men. This phenomenon of male aging being associated with increased ROS levels is already extensively studies in the literature and is also referenced in the manuscript. Therefore we did not include a younger group of men in our study.
Comment 2: Conceptionally, there is an issue to portrait idebenone as an antioxidant only (see current literature). Especially given the very small effect on oxidative stress levels in sperm.
Response: We agree with the reviewer that idebenone has been recently shown to work on multiple pathways outside of its antioxidant effects, including activation of RNA binding proteins (i.e. Lin28A), inhibition of p52Shc, increasing ATP levels, reduce inflammation and reduce ER stress (doi.org/10.1016/j.redox.2020.101812). We have broadened out introduction to include other potential benefits of idebenone and added a further paragraph to the discussion about how idebenone maybe have improved embryo outcomes outside its antioxidant properties.
Introduction = ‘More recently idebenone has been shown to shown to display molecular activity outside of an antioxidant including; bioactivity of idebenone metabolites, protein inhibition (i.e. p52Shc), regulation of gene transcription (i.e. Lin24A) and reductions in inflammation and endoplasmic reticulum stress [42].’
Discussion = ‘Recently, idebenone has been shown to have additional molecular activity outside that of oxidative damage protection. These include [44]; improved mitochondrial respiration (as discussed above), selective PPARα/γ agonist, inhibition of p52Shc which acts as an adaptor protein required for a variety of molecular complexes most notability protein tyrosine kinase receptors (which are vital in sperm capacitation, acrosomal exocytosis and gamete fusion [83]), upregulation of Lin28A (although unlikely to be of importance to sperm, due to lack of transcription), reductions to inflammation (commonly seen in aging [84]) and reductions to endoplasmic reticulum stress (again commonly seen in aging [85]). Therefore, improvements to some of these other molecular pathways of aged sperm by idebenone could also be contributing to the benefits seen on downstream embryo development and health.’
Comment 3: It is well known and described in reference 41 that idebenone, to act as antioxidant, has to be bioactivated (reduced). Do the authors know about the presence of a bioactivating system in sperm? If so, why was this not shown/described/cited? If not, the study approach lacks a major justification. The cited references used isolated mitochondria supplied with substrates to fuel a complex II dependent bioactivation. Is not what happens in intact cells.
Response: Sperm contain the NAD(P)H quinone oxidoreductase (NQO also known as DT- diaphorase PMID: 1520038, 16595077) and a sperm specific enzyme with diaphorase activity (Science 191:1185-1187). It is therefore highly likely that bioactivation of idebebone can occur within in sperm. We apologise for this omission in the original manuscript. We have added the following to the discussion of the manuscript.
Discussion = ‘In addition to its scavenging antioxidant properties, idebenone also acts as an electron transporter once reduced to its hydroquinone form, idebenol, by NAD(P)H quinone oxidoreductase 1 (NQO1) [67]. Sperm contain NAD(P)H quinone oxidoreductase (NQO also known as DT- diaphorase [68,69]) and a sperm specific enzyme with diaphorase activity [70], suggesting that bioactivation of idebebone can occur within in sperm.
Comment 4: The bioactivation of idebenone also reduces the latent pro-oxidative activity of idebenone, this feature was not discussed although it could explain the lack of dose effect observed in Figure 1 that the authors address in the discussion.
Response: If the reviewer refers to lines 459-469 of the discussion, they will find that the latent pro-oxidative activity of idebenone was in fact discussed in relation to the lack of a dose effect:
‘This study has also highlighted an important factor that may be contributing to the limited clinical uptake of antioxidants in human sperm culture media, which is that excessive amounts of antioxidants can interfere with physiological ROS concentrations, leading to enhanced ROS generation in mitochondria and further oxidative injury to cells [84,85]. In this study 50µM of idebenone had no effect on sperm superoxide levels in both human and mouse. Previous studies have shown that idebenone can act as a pro-oxidant through the inhibition of complex 1 of the electron transport chain, thereby promoting superoxide production [86-88], with concentrations >25µM causing apoptosis in human neuronal cell lines in vitro [89]. Therefore, its pro-oxidant properties coupled with a higher concentration (i.e. 50µM) may have the opposite effect, leading to a state of reductive stress and no change to sperm ROS concentrations [84,85].’
Comment 5: It seems that the authors conceptually used idebenone as a simple antioxidant and did not even contemplate the notion that the observed effects for example on blastocyst survival and implantation rate could be caused by large number of additional effects unrelated to antioxidant effects. See for example Lei et al. 2018
Response: Please refer to response to comment 2. We have added a more extensive explanation of the fact that idebenone may also act by other pathways outside of its antioxidant effect to the introduction and discussion.
Comment 6: If idebenone is supposed to be portrayed as a “better” antioxidant (line 90), why was no control group included to show its superiority?
Response: We have modified our introduction to better prove our point for why we choose to use idebenone. This includes a list of positive attributes making it more suitable for clinical ART use;
- Mitochondrial permeable
-Water soluble
- Already been approved for use treat Duchenne muscular dystrophy.
Introduction = ‘It shares similarities with the antioxidant ubiquinone, known as co-enzyme Q10, however is more readily soluble in gamete/embryo-compatible culture media solutions, is able to permeate cellular membranes in vitro and is already been approved by the Therapeutics Goods Administrator (TGA) and US Food and Drug Administration (FDA) to treat Duchenne muscular dystrophy [42], thus making it a good candidate for use in clinical ART.’
Based on our results and conclusion of the study you can see that we have not indicated that idebenone is in fact the best choice, but further work is still required.
Conclusion = ‘Further studies to determine the optimal doses, best antioxidant types, exposure length to sperm in vitro, understanding those patients who would benefit the most and downstream consequences to embryo quality and pregnancy outcomes in humans are still required to fully understand whether or not it’s plausible and clinically safe to add antioxidants to sperm preparation media.’
Comment 7: The antioxidant effect of idebenone was previously predominantly shown as a reduction of lipid peroxidation. It is therefore unexplained why this endpoint was not assessed (by using for example BODIPY-C11 dye) but instead water-soluble ROS dyes such as DCFDA were used. Could this be the reason why no major antioxidant effects were observed? As an alternative to measuring ROS, measuring ROS-induced damage (i.e. lipid peroxidation, protein-nitrosylation or oxidative DNA damage via 8-oxo-dG) could have been more informative.
Response: We chose to measure superoxide production (MitoSOX red) and hydrogen peroxide and hydroxyl radicals (DCFDA) over lipid peroxidation in our samples as these are upstream in the lipid peroxidative pathway and are instrumental in the formation of lipid peroxidation (PMID: 18846257). Given that our treatment was only for 1 h, we wanted to measure upstream markers to determine if idebenone was influencing ROS concentrations in sperm.
While we acknowledge that measuring some additional ROS-induced damage markers as suggested, would have been informative we did not assess 8-oxo-dG levels in the mouse due to their low baseline levels (PMID: 26510519) due to their very high protamine content.
Comment 8: Protection against pro-oxidative damage (i.e. DNA damage) could have supported the presence of an antioxidant effect. However, it is unclear why DNA damage was not directly assessed. In contrast caspase 3 staining was employed. The caspase staining in Figure 2k is not convincing, it even shows a higher signal with idebenone (it is unclear what we are looking at in this image, what are the bright spots and what exactly was quantified, how about a counterstain with DAPI to get an orientation?). DNA damage could have been much more directly and convincingly demonstrated using comet assays or gammaH2AX staining. It could have also answered the question if the DNA damage in the blastocysts was inherited from the sperm or originated in the blastocysts.
Response: We have made it clearer in the methods how Caspase 3 was quantified and have separated out the figure to show the individual nuclear and Caspase 3 channels, which we believe will help to orientate readers to those cells that were positive to Caspase 3 in the blastocyst (see re worked figure 2k).
Methods = ‘For cells to be classified as apoptotic Caspase-3 (green) had to co-localize with nuclear (Hoechst – blue) staining’.
We assessed Caspase 3 a marker of apoptosis for two reasons. (1) It had been previously shown to be reduced by idebenone treatment in other cell types (PMID: 26284974, 30559635) and (2) given the increase in cell numbers of our blastocyst due to idebenone treatment we wanted to determine if these improvements were due to a decrease in cellular apoptosis or as it suggests based on our data likely from increased cell division. We have modified the discussion to reflect this.
Discussion = ‘idebenone has been shown to suppress levels of active Caspase-3 in human umbilical cord vascular endothelial cells [43] and rat livers [82], and therefore was selected for use as a marker of apoptosis in this study. Whilst the relationship between sperm oxidative stress and apoptosis in the blastocyst had been previously shown [31], our study did not show a reduction in active Caspase-3 positive cells in blastocysts produced from aged sperm with or without 5µM idebenone. A limitation of measuring active Caspase-3 in blastocysts is that it may not be reflective of the sperm DNA damage inherited by the embryo which maybe better detected by gammaH2AX staining or the comet assay.’
Discussion = ‘However, given that blastocyst cell numbers were increased in embryos produced from idebenone treated sperm, suggests that this increase in cell numbers was more likely due to an increase in cellular division during embryogenesis and not due to a reduction in cellular apoptosis.’
Comment 9: The discussion on the properties of idebenone is too simplistic at several levels: for example the interaction with complex III of the mitochondria (line 398) was only ever demonstrated when complex I was impaired. Is there any evidence that this happens in aged sperm? What about all the other effects idebenone has on cells?
Response: We have modified our discussion to better indicate that aged sperm contain phenotypes (i.e. reduced sperm motility and altered mitochondrial function PMID:20021411, doi.org/10.1016/j.jevs.2016.10.015), that would indicated impairments to the electron transport chain of their mitochondria.
Discussion = ‘Idebenone can increase/restore ATP levels under stress conditions with impairments to mitochondrial complex 1 function, by shuttling electrons to complex III of the electron transport chain [63,75]. Mitochondrial energy metabolism is crucial for male reproductive function, with aging associated with reduced mitochondrial respiration rates of sperm [76,77] and decreased sperm motility. Thus idebenone may be increasing the motility and fertilizing capacity of sperm from older males [9], potentially via improvements to mitochondrial function and ATP production, however furthers studies are required to test this.’
Comment 10: The study does not provide any evidence of a causative nature that antioxidant function (if indeed present) is responsible for improved embryo quality. Therefore, it is descriptive and as such should be discussed in this manner.
Response: We have modified our conclusion as suggested to be more of a descriptive nature. However, we do continue to believe that idebenone still has an antioxidant function as evident by the reductions to superoxide production of sperm seen in figure 1 at 5µM and the fact that sperm contain the enzyme required for idebenone bio-activation.
Comment 11: Given the extent of literature on antioxidants (or compounds presumed to be antioxidants) and sperm quality, I do not think that the concluding statement is adequate and should be reworded.
Response: We acknowledge the large body of work that has previously looked at the addition of antioxidants to sperm both in fresh and frozen samples and the reviewers point. We have modified the conclusion slightly to include downstream outcomes to embryo quality and pregnancy outcomes. However we stand by our original conclusion, because if the current literature was in fact ‘adequate’ we would expect that the addition of antioxidants to clinical IVF media to be main stream, which it is not. Currently, there is only one company (Vitrolife) that we know of that is looking into the addition of antioxidants to its embryo culture media (ACTRN12618001479291) which is still in clinical trial phase.
Minor points:
Comment 1: How were ROS levels quantified in Figure 1? Which peak was used for quantification?
Response: ROS levels were quantified by mean fluorescent intensity (fluorescent units) which we have made clearer in the methods. We have also re added the gating’s back to our histograms representative pictures so you can see which peaks were used for quantification. We had originally removed these due to overcrowding on the image.
Comment 2: The references for the antioxidant effect of idebenone (42, 47) are really old and should be replaced with contemporary references that are suited to support the statements. Especially since both references refer to cell free studies and are therefore unsuited to justify the doses used with intact cells. Again, the bioactivation process must be accounted for.
Response: As suggested by reviewer 1, we have added in additional references to this statement to support the notion that idebenone has antioxidant capabilities in a variety of cells.
Introduction = ‘In vitro, idebenone has already been shown to reduce ROS concentrations and cell death in retinal epithelium [42], lipid peroxidation of vascular endothelial cells [43], reduced ROS formation in rat brain synaptosomes [44] and apoptotic cell death in optic nerve astrocytes, all which are a hallmark features of oxidative damage and common phenotype seen in sperm from men of an advanced aged [45].’
Comment 3: The figure legends do not illustrate, which statistical tests were used for each figure. This makes the interpretation of the validity of the results impossible. For example, I struggle to see how Figure 2b can show a statistically significant effect based on the individual samples shown as dots. Illustration of how the samples from each donor reacted by connecting the relevant sample dots with lines might illustrate this effect better.
Response: As suggested by the reviewer we have modified our graphs, so each individual biological replicate has a different symbol shape (please see figure 1 and 2). This will hopefully help the readers to track individual males/animals across treatments and see the individual shifts due to idebenone exposure. We have also adding the statistical tests performed to the figure legends for each measure also. For instance, the statistical test analyzed for Fig 2b was a paired t-test.
Comment 4: Line 91: vitamins E and C are cell permeable and especially vitamin C as an acid is very water soluble. Therefore, the statement that they are ALL lipophilic is plain wrong. While I fully agree that CoQ10 is unsuited for the purpose of this study as in vitro it takes at least 9 days to reach the mitochondria, vitamin C and E will reach them in a few minutes.
Response: We would like to thank the reviewer for pointing out this mistake in our introduction. Vitamin C should have not been included in this sentence which we have now removed. However, we have left the statement regarding CoQ10 and Vitamin E as is as they are very lipophilic, they tend to be retained in cell membranes, and fail to achieve significant intracellular concentrations, which is why mitochondrial permeable versions of these have been established (MitoQ and MitoVitE) (PMID: 17025271).
Comment 5: To how much idebenone were the oocytes possibly exposed to during the procedure?
Response: Oocytes would have only been exposed to a minimal/limited amount of idebenone during the IVF insemination (~4h) from a small carry over from sperm. As sperm are inseminated at a 1x105/ml of sperm, with sperm extracted from mice varying between 8-30x106/ml we would estimate oocytes would have been exposed to between ~0.07-0.01µM of idebenone during this time period depending on biological sample. Post IVF, all oocytes are washed multiple times to remove the G-IVF prior to been put in G1 culture and therefore potential exposure is only limited during that window.
Reviewer 2 Report
In the manuscript titled “Improving sperm oxidative stress and embryo quality in advanced paternal age using idebenone in vitro – a proof of concept study.” the authors analyse idebenone as antioxidant in order to improve sperm oxidative stress and embryo quality in advanced paternal age.
I thing that this work was carefully conducted. It should be integrated considering the following aspects:
- Line 20: delete a “fertilization”; it is written two times.
- In the abstract some sentences are too long.
- Why the concentration 5µM and 50µM Idebenone were used?
- Why the treatment was for 1 h?
- The authors state that both 5µM and 50µM of idebenone to mouse semen were able to reduce hydrogen peroxide and hydroxyl radicals (DCFDA), however the same was not seen in humans. What explanation do the authors give for this difference?
- the authors should better explain the molecular mechanism by which idebenone produces these effects.
- do the authors have any idea of the status of protamines in the sperm of these subjects?
- Is there a canonical ratio of protamines to sperm? or not? The canonical protamine/histone ratio is crucial for protecting DNA from oxidative damage. DNA fragmentation is one of the main causes of infertility, but a non-canonical protamine/histone ratio can lead to increased DNA fragmentation. This aspect must also be considered in the discussion. In this regard, I recommend reading and quoting the following article (PMID: 32545547)
Author Response
Reviewer 2
In the manuscript titled “Improving sperm oxidative stress and embryo quality in advanced paternal age using idebenone in vitro – a proof of concept study.” the authors analyses idebenone as antioxidant in order to improve sperm oxidative stress and embryo quality in advanced paternal age. I think that this work was carefully conducted. It should be integrated considering the following aspects:
Comment 1: Line 20: delete a “fertilization”; it is written two times.
Response: We have deleted the duplication of the word fertilization from line 20 of the abstract.
Comment 2: In the abstract some sentences are too long.
Response: We have modified the abstract to reduce the length of several of the sentences as per the reviewer suggestion.
Comment 3: Why the concentration 5µM and 50µM Idebenone were used?
Response: We have made it clearer in the methods that these concentrations of idebenone were chosen based on previous literature.
‘G-IVF PLUS medium (Vitrolife, Denver, Colorado, USA) was supplemented with either 5µM or 50µM of idebenone, with these concentrations previously been shown to reduce ROS related cellular damage up to 50% in brain cells and prevent ROS related cell death in retinal epithelium [42,44,48,49]’.
Comment 4: Why the treatment was for 1 h?
Reponses: A 1h incubation was chosen as this approximates the average length of time for a traditional swim up (sperm preparation method) performed in clinical IVF. We have made this clearer in the methods section of the manuscript.
Comment 5: The authors state that both 5µM and 50µM of idebenone to mouse semen were able to reduce hydrogen peroxide and hydroxyl radicals (DCFDA), however the same was not seen in humans. What explanation do the authors give for this difference?
Response: We believe that the likely reason why mouse sperm were able to reduce both the hydrogen peroxide and hydroxyl radicals, while the same was not seen in humans was likely due to the reduced biological and lifestyle variability of research rodents. The mice used while outbreed, were kept in a very controlled research environment limiting other variables (i.e. nutrition) that could modify baseline ROS concentrations, resulting in more consistent results. Although, men in this cohort were of an advanced age (>40 years) other biological and lifestyle factors (not obtained) could have resulted in some men having higher baseline ROS concentrations, which may require a higher concentration (>50µM) of idebenone to illicit a reduction in sperm ROS. We have added this paragraph to the discussion to try and explain this difference to the reader.
Comment 6: the authors should better explain the molecular mechanism by which idebenone produces these effects.
Response: Similar to reviewer 1 comments, we have provided further details in both the introduction and in the discussion, for the idebenone mode of action and how it may be eliciting its positive effects to sperm and early embryo health.
Comment 7: do the authors have any idea of the status of protamines in the sperm of these subjects?
Response: Unfortunately, beside participants age, BMI and that they were normospermic, no other fertility information was obtained from our donors and their protamine status was unknown. We do believe that it would be interesting in the future to look at the interplay between sperm ROS concentrations and protamine levels, given the higher abundance of histone retention in men undergoing infertility treatment (PMID: 21685136).
Comment 8: Is there a canonical ratio of protamines to sperm? or not? The canonical protamine/histone ratio is crucial for protecting DNA from oxidative damage. DNA fragmentation is one of the main causes of infertility, but a non-canonical protamine/histone ratio can lead to increased DNA fragmentation. This aspect must also be considered in the discussion. In this regard, I recommend reading and quoting the following article (PMID: 32545547).
Response: Thank you for your suggestion and we enjoyed reading the suggested article about nuclear base proteins on DNA oxidative damage in healthy males presenting copper and chromium excess in their semen. However, as we did not directly measure oxidative DNA damage in our samples or histone to protamine ratios, we do not want to overstate/speculate our findings in the discussion (as indicated by reviewer 1) in relation to idebenone treatment and therefore, have not added this to our discussion.
Round 2
Reviewer 2 Report
accept in the present form